# The Relationship between Perceived Social Support and Life Satisfaction: The Chain Mediating Effect of Resilience and Depression among Chinese Medical Staff

**DOI:** 10.3390/ijerph192416646

**Published:** 2022-12-11

**Authors:** Nannan Wu, Fan Ding, Ronghua Zhang, Yaoyao Cai, Hongfei Zhang

**Affiliations:** 1Mental Health Education and Counseling Center, Shaoguan University, Shaoguan 512005, China; 2School of Intelligence Engineering, Shaoguan University, Shaoguan 512005, China; 3Institute of Developmental and Educational Psychology, School of Marxism, Wuhan University, Wuhan 430072, China

**Keywords:** perceived social support, life satisfaction, resilience, depression, Chinese medical staff

## Abstract

Medical staff are direct providers of medical services and a key element in the development of health services, and their life satisfaction is important to both their work satisfaction and their patients’ satisfaction, subsequently influencing the quality of medical care in general. This cross-sectional study aimed to explore the mechanisms underlying the influence of perceived social support on medical staff’s life satisfaction. Convenience sampling was used to recruit participants from two non-tertiary hospitals in Shaoguan City, Guandong Province, China. A total of 533 medical staff completed the Multidimensional Scale of Perceived Social Support, the Satisfaction with Life Scale, the Connor and Davidson Resilience Scale, and the depression subscale of the Depression, Anxiety, and Stress Scales (DASS-21). The results showed that perceived social support could influence medical staff’s life satisfaction not only through the separate effects of resilience and depression, but also through the chain mediation effect of resilience and depression. This study suggests that reducing the depressive symptoms of medical staff and improving their perceived social support as well as resilience could help to enhance their life satisfaction.

## 1. Introduction

Life satisfaction refers to individuals’ global cognitive evaluation of the extent to which they are satisfied with their own lives [1]. As a key component in the attainment of positive mental health, life satisfaction is closely associated with a broad range of behavioral, psychological, social, interpersonal, and intrapersonal outcomes, such as stress mitigation and the reduction of externalizing behaviors and problem internalization [2]. Thus, life satisfaction is considered a key indicator of people’s subjective wellbeing as well as quality of life [3].

In recent years, life satisfaction has been regarded as a crucial indicator of medical staff’s wellness and has drawn increasing scientific attention as a significant public health issue around the world [4]. Many previous studies have revealed that medical workers’ life satisfaction is related to personal factors (e.g., age, gender, professional achievements, personal satisfaction, etc. [5,6]) and contextual factors (working environment-related factors, such as workload, relationships with colleagues, payment, income, etc. [5,6,7]), as well as intrinsic characteristics of the medical profession (e.g., patient interactions, demographics and complexity, as well as the staff member’s own specialty [5,8]). In particular, a great many of these studies have demonstrated that medical professionals are prone to exposure to lots of psychosocial risk factors due to the special nature of this profession, such as the high intensity of the workload, relatively long working hours, violent, abusive, demanding patients, and so on [5,6,7,8,9]. This is alarming, because the aforementioned risk factors are not only related to the medical staff’s job and life satisfaction, but are also associated with the quality of medical care, which may further effect the general relationship between physicians and patients [10]. Especially during the worldwide outbreak of the Coronavirus disease in 2019, medical providers’ physical and mental health were indispensable for their productivity and effectiveness during medical operations [11]. Therefore, it is important to pay more attention to the mental health and life satisfaction of medical staff.

In developed countries, medical workers usually have a high social status and high income [12]. However, as a densely populated developing country, China has a cohort of medical providers who are faced with a rather different and more difficult situation. For example, at the end of 2019, the number of physicians in China was 2.77 per 1000 population [13], whereas the number of physicians per 1000 population was 4.3 in the Czech Republic, 4.0 in Italy, and 4.2 in Germany at the end of 2016 [4,13], which means that medical workers in China are faced with heavier workloads than their counterparts in developed countries [14]. Furthermore, in the process of healthcare reform, Chinese physicians have been required to take on more responsibilities for patient care, which in turn has resulted in an increase in their workloads [15]. Considering the scarcity of medical providers and their work burden and intensity as compared to developed countries, we believe that Chinese medical staff may be at a higher risk of psychological problems.

Above all, medical staff’s life satisfaction is an important public health and social problem all over the world, especially in China [16]. Therefore, this study aims to explore the factors that influence medical staff’s life satisfaction literacy as well as its underlying mechanisms, so as to lay a solid foundation for the development of more complete plans for improving their life satisfaction.

### 1.1. Perceived Social Support and Life Satisfaction

Social support refers to an individual’s belief in the existence of support from family, friends, and significant others in his/her life [17]. It is an important and effective psychological resource that allows individuals to deal with psychological tension, protect them from stressful and oppressive events, enhance their social adaptability, and make them more resilient to adverse conditions. In general, social support is composed of both received social support and perceived social support. Received social support is the objective support coming from others near to individuals, which focuses on the quantity and quality of the given support, whereas perceived social support reflects the perceived availability and adequacy of social connections, referring to the subjective perception and assessment of support originating from family, friends, and significant others [18,19]. Compared to received social support, perceived social support provides a more important and effective assessment of an individual’s mental health [20,21]. Therefore, combining the above arguments in the literature [18,19,20,21], the current study adopted perceived social support rather than received social support as an independent variable, investigating how perceived social support exerts an effect on an individual’s life satisfaction.

According to the main effect model of social support, social support is beneficial for mental health, eliciting an increased sense of wellbeing and life satisfaction [22]. In recent years, perceived social support has been confirmed to be an important indicator of life satisfaction in a wide variety of population groups, such as the elderly [23], adolescents [24], pregnant women [25], individuals with substance use disorder [21], medical social workers [26], and so on. These previous studies have confirmed that perceived social support was positively associated with people’s life satisfaction, meaning that the higher the perceived social support, the higher one’s life satisfaction [23,24,25,26]. Conversely, a lack of perceived social support from family, friends, and significant others has widely proved to decrease one’s sense of satisfaction in life [24,25]. Although the influence of perceived social support on life satisfaction has been well-established, the potential mechanisms accounting for this correlation remain unclear due to its complexity [27]. In addition, little attention is paid towards the relationship between perceived social support and life satisfaction among medical staff, especially those in China. Therefore, this study attempts to explore the potential mechanisms in the link between perceived social support and life satisfaction in a sample of Chinese medical staff.

### 1.2. Perceived Social Support, Resilience, and Life Satisfaction

Psychological resilience is regarded as an important part of positive psychology, which believes that resilience is a dynamic process [28]. As an important psychological resource, resilience is usually defined as the ability to “bounce back” from stressful experiences quickly and effectively, to adapt to and grow in response to volatile situations, such as crises, work burdens, trauma, and other adversities [29]. Thus, in unfavorable situations, highly resilient individuals easily adapt to and recover from setbacks, whereas less resilient individuals are less likely to adapt to and recover from stressful experiences [30].

Both theoretical and empirical studies have explored the associations among perceived social support, resilience, and life satisfaction. Resilience could be generated from relational and social factors such as family bonds and supportive relationships [31]. Thus, perceived social support is widely regarded as an external protective factor for resilience, which has been confirmed by many previous studies [28,31,32]. Moreover, perceived social support has been indicated to be positively correlated with resilience [33]. Zhang et al. suggested that perceived social support was significantly positively correlated with resilience among Chinese college students, since students with high levels of perceived social support were reported to have high levels of resilience [34]. Several other studies have also investigated the relationship between resilience and life satisfaction and proved that increasing resilience predicted an increase in life satisfaction [35]. This is because individuals with greater levels of resilience can use appropriate emotional and psychological resources to recover from negative experiences, maintain happiness in the face of adversity, and then continue to pursue and achieve goals, and they are thus able to experience higher life satisfaction [32,36]. In addition, resilience has been shown to serve as an important mediating role between perceived social support and life satisfaction. For example, Guo et al. showed that resilience could enhance the protective effects of perceived social support on college students’ life satisfaction [35]. A more recent study by Yang et al. also revealed that resilience significantly mediated the relationship between perceived social support and life satisfaction among individuals with substance use disorders [21]. The above precedents inspired the current study. Because perceived social support is related to resilience, and both resilience as well as perceived social support are predictors of life satisfaction, the possible mediating effect of resilience on the link between perceived social support and life satisfaction should be assessed to reinforce our understanding of the mental health of medical staff. However, studies investigating the mediating role that resilience plays are still insufficient, especially among medical providers in China’s non-tertiary hospitals.

### 1.3. Perceived Social Support, Resilience, Depression, and Life Satisfaction

Depression is recognized to be one of the most common mental disorders and refers to a state of negative emotion, such as sadness, helpless, loneliness, poor appetite, drowsiness, and so on [37]. A cross-sectional study by Muñoz-Bermejo et al. confirmed that perceiving social support was important in preventing the appearance of depressive symptoms [38]. Similarly, Tariq et al. indicated that perceived social support had a significant predictive effect on depressive symptoms in the elderly of Pakistan, with depression worsening as the level of perceived social support decreased [39]. Furthermore, evidence from previous studies has indicated that there is a significantly negative relationship between depression and life satisfaction [40], meaning that less depressed people have higher life satisfaction [41]. Thus, depression is negatively associated with perceived social support as well as life satisfaction. In addition, several studies have indicated that individuals with higher levels of perceived social support were at lower risks for negative mental health consequences, such as social pressure, anxiety, and depression, and they would thus experience higher levels satisfaction of life [42]. According to the correlations mentioned above, depression may be inferred to be a mediator in the relationship between perceived social support and life satisfaction.

Previous research has found that resilience protects people from developing serious mental disorders, including depression [43]. Specifically, highly resilient individuals usually demonstrate stronger beliefs in overcoming stresses and thus tend to be at less of a risk for depression [44]. Therefore, psychological resilience, to some degree, could prevent the occurrence of depression in an individual [45]. Furthermore, resilience has been indicated to play an important mediating role between the relationships among depression and several psychological variables, e.g., perceived social support [40], life satisfaction [46], and adaptability to negative events [47]. Based on all the literature we consulted on the associations among perceived social support, resilience, depression, and life satisfaction, this study assumes that perceived social support may exert an indirect effect on life satisfaction through the mediating chain between resilience and depression.

### 1.4. The Current Study

The above literature has suggested the intertwined and cyclical associations between perceived social support, resilience, and depression, all of which have been proven to be essential factors in maintaining life satisfaction. However, as far as we know, there is very scarce available literature simultaneously investigating the associations among all four of the variables. Furthermore, previous studies have mostly focused on the utilization of multiple logistics regression to investigate the influential factors in life satisfaction in medical workers, and these have hardly dived into the internal relationships between variables. By contrast, by analyzing mediating effects, we are able to demonstrate and explain how variables directly or indirectly interact with each other [46]. Based on a sample of Chinese medical staff, our study attempts to demonstrate the underlying relationships among perceived social support, resilience, depression, and life satisfaction, as well as to examine the chain mediating role of resilience and depression in the impact of perceived social support on life satisfaction. To this end, based on previous studies, we propose the following four hypotheses: (1) Perceived social support is positively associated with medical staff’s life satisfaction. (2) Resilience mediates the relationship between perceived social support and medical staff’s life satisfaction. (3) Depression may mediate the relationship between perceived social support and medical staff’s life satisfaction. (4) Resilience and depression may have a chain mediating effect in the relationship between perceived social support and medical staff’s life satisfaction.

## 2. Materials and Methods

### 2.1. Participants

This cross-sectional study was conducted with feedback from online self-reported questionnaires completed by medical providers in Shaoguan, China from 5 July 2022 to 25 July 2022. Participants from 2 non-tertiary hospitals (including a maternal and child healthcare hospital and a general purpose hospital) in Shaoguan were recruited through convenience sampling. Specifically, medical providers were invited to participate in this study if they were: (1) employees with more than one year’s working experience, (2) without past diagnosis of psychiatric illnesses and/or family history of psychosis, and (3) without any use of psychiatric medicines.

### 2.2. Data Collection

The questionnaire was delivered through a professional questionnaire survey platform entity, “*Wenjuanxing*” (www.wjx.cn. (accessed on 5 July 2022)), and then distributed through social media with the WeChat platform. *Wenjuanxing* is a popular social media-based survey platform whereby researchers are able to communicate research findings and academic progress with each other [48]. While designing the online questionnaire, we employed this platform’s data integrity check function to ensure that all questions within the questionnaire were properly answered before submission.

In order to further ensure the quality of our survey, questionnaires completed and submitted within 5 min or more than 30 min were excluded, and we made sure that a single IP corresponded to one submission. In addition, questionnaires with obviously inappropriate response patterns, such as those featuring the same response to all questions, were discarded.

### 2.3. Measurements

#### 2.3.1. Sociodemographic Characteristics

The questionnaire collected data on Chinese medical staff’s sociodemographic information, such as age, gender, marital status (married or single), education level (college diploma or bachelor’s degree or higher), and other job-related factors including education level, professional titles (primary, intermediate, senior), working experience in years, monthly income, and so on.

#### 2.3.2. Perceived Social Support

The Multidimensional Scale of Perceived Social Support (MSPSS) was utilized to measure perceived social support from family, friends, and significant others. The MSPSS scale was developed by Zimet et al. [49] and translated into Chinese by Jiang [50]. Further, this scale includes 12 items across 3 dimensions: family support, the support of friend, and the support of a significant other. Participants rated each item on a seven-point Likert scale (from 1 = very strongly disagree to 7 = very strongly agree). The mean total score for the MSPSS scale ranged from 1 to 7 points, with higher scores indicating higher perceived social support. In this study, Cronbach’s alpha of the MSPSS was 0.945.

#### 2.3.3. Life Satisfaction

The Satisfaction with Life Scale (SWLS) developed by Diener et al. [3] was used to assess life satisfaction. The SWLS consists of 5 items, with each item on a Likert-type scale ranging from 1 (strongly disagree) to 7 (strongly agree). The total scale score of life satisfaction was determined by averaging the 5 items in the current study, which ranged from 1 to 7 points. The SWLS has been widely used in the research on people’s quality of life in China and was found to be reliable [9]. Cronbach’s alpha of the scale was 0.917 in this research.

#### 2.3.4. Resilience

The Chinese version of the Connor–Davidson Resilience Scale (CD-RISC) was used to assess resilience, which was compiled by Connor and Davidson [51] and translated by Yu et al. in 2007 [52].The Chinese version of the CD-RISC includes 25 items across 3 dimensions: tenacity, strength, and optimism, with responses on a five-point Likert scale from 0 (“never”) to 4 (“always this”). The mean for the total scale ranged from 0 to 4 points, with higher scores implying a higher level of resilience. Cronbach’s alpha of the CD-RISC in this study was 0.958.

#### 2.3.5. Depression

The 21-item Depression, Anxiety, and Stress Scales (DASS-21) is a compilation of three subscales of a self-reporting questionnaire intended to evaluate the mental states of depression, anxiety, and stress, respectively. Each subscale contains 7 items, and there are four options used to answer each item, ranging from 0 (not applicable) to 3 (mostly applicable). Note that one unique feature of the DASS-21 Depression (DASS-D) subscale lies in the fact that it contains no somatic items, which makes the DASS-D subscale an ideal tool to evaluate depression in individuals affected by pain [53], such as fatigue, sleep disturbance, and so on. Moreover, previous studies have revealed that high-quality evidence of sufficient criterion validity was exhibited for the DASS-D subscale [54]. Therefore, the 7-item DASS-D subscale translated by Yi et al. [55] was chosen to evaluate participants’ depressive symptoms in this study, due to the aforementioned merits as well as its easy implementation. Moreover, scores on the DASS-D subscale were multiplied by 2 [54,55], so that the mean score ranged from 0 to 6 points, with higher scores signifying higher depressive symptoms. In the present survey, Cronbach’s alpha for the DASS-D subscale was 0.885 in the Chinese medical staff group.

### 2.4. Statistical Analysis

IBM SPSS Statistics 26.0 was utilized to analyze the data collected in the current study. Continuous variables are represented by mean ± standard deviation (SD), and categorical variables are represented by frequencies as well as percentages. Before proceeding to the analysis, the normality of the four main variables was tested with the SPSS program. The resulting histograms and P-P plots for the mean scores of perceived social support, resilience, and life satisfaction displayed an approximately normal distribution, with skewness and kurtosis values within approximately ±0.5 and ±0.1, respectively. For the mean score of depression measured by DASS-D, the values of skewness and kurtosis were 1.47 and 2.64, respectively, which were also in the acceptable range (skewness values <|3| and kurtosis values <|10|, [56,57]), and thus the depression data exhibited roughly a normal distribution. Moreover, parametric tests that use the *t* (*t*-test) and *F*-statistics (ANOVA) are generally robust for violations of normality as long as group sizes are relatively large [58]. Therefore, in our analysis of the main variables in relation to different socio-demographic variables, an independent samples t-test and a one-way ANOVA were used. Subsequently, Pearson’s bivariate correlation analysis was conducted to analyze the associations between measures. In addition, Model 6 in PROCESS macro compiled by Hayes [59] was employed to test the significance of the chain mediation effect. A Bootstrap 95% confidence interval (CI) was adopted to assess whether the regression coefficients were significant for estimating the chain mediation effect from 5000 samples in the original data using repeated random sampling. If the 95% CI did not contain 0, the indirect effect was considered to be statistically significant.

## 3. Results

### 3.1. Common Method Bias Test

Since the questionnaire approach employed in this study may result in common method deviation, Harman’s single-factor method was used for incorporating perceived social support, resilience, depression, and life satisfaction items in an exploratory factor analysis. The results showed that there were seven factors with eigenvalues greater than 1, and the variation explained by the first factor was 26.7%, which was less than the critical value of 40%. Therefore, there was no significant common method deviation.

### 3.2. Socio-Demographic Characteristics

A total of 533 questionnaires were collected, and 491 valid questionnaires were recovered, with an effective recovery rate of 92.12%. Of the 491 participants, 82.1% (*n* = 403) were females, 70.7% (*n* = 347) were married, and 61.3% (*n* = 301) were formal employees. Their ages ranged from 20 to 60 years old, with an average age of 35.11 (*SD* = 10.18). Moreover, 56.6% (*n* = 278) of the participants had received a college diploma, and 43.4% (*n* = 213) had a bachelor’s degree. Professionally, 30.6% (*n* = 150) of them were doctors, 52.7% (*n* = 259) were nurses, and 58.9% (*n* = 289) had a primary professional title. Moreover, almost half of the participants (54.4%, *n* = 267) had monthly incomes ranging from RMB 3000 to RMB 6000, and 48.5% (*n* = 238) of them had working experience of less than 10 years. More detailed socio-demographic characteristics of the medical workers who participated in this study are summarized in Table 1.

### 3.3. Correlation Analysis

Table 2 presents the mean, standard deviations (SD), and Pearson correlation coefficients between the major variables. Perceived social support was significantly positively correlated with Chinese medical staff’s life satisfaction (*r* = 0.659, *p* < 0.01) as well as resilience (*r* = 0.566, *p* < 0.01), whereas it was negatively related with depression (*r* = −0.589, *p* < 0.01). Moreover, resilience was observed to be negatively related with depression (*r* = −0.574, *p* < 0.01), but positively associated with life satisfaction (*r* = 0.571, *p* < 0.01). Finally, depression was significantly negatively correlated with Chinese medical staff’s life satisfaction (*r* = −0.633, *p* < 0.01).

### 3.4. Chain Mediation Effect Test

Utilizing Model 6 in the PROCESS macro developed by Hayes, a 95% confidence interval (CI) of the chain mediation effect of resilience and depression on the impact of perceived social support on Chinese medical staff’s life satisfaction was evaluated, and the chain mediation model was established as shown in Figure 1. Moreover, age, work year, professional title, monthly income, marital status, and position were set as covariates in this study. The regression results presented in Table 3 showed that the direct predictive effect of perceived social support on life satisfaction was significant and positive (*β* = 0.447, *p* < 0.001). Moreover, perceived social support could significantly and positively predict resilience (*β* = 0.384, *p* < 0.001) and negatively predict depression (*β* = −0.373, *p* < 0.001). In addition, resilience could significantly negatively predict depression (*β* = −0.489, *p* < 0.001), whereas it could significantly positively predict life satisfaction (*β* = 0.351, *p* < 0.001). Finally, depression could negatively predict life satisfaction (*β* = −0.397, *p* < 0.001).

The bootstrapping method was used to resample 5,000 times to calculate for a 95% CI. As shown in Table 4, the results showed that psychological resilience and depression play an intermediary role between perceived social support and life satisfaction, and the total mediating effect was 0.358 (95% CI = (0.279, 0.442)), accounting for 44.47% of the total effect. The indirect effect on the perceived social support → resilience → life satisfaction path was 0.135 (95% CI = (0.07, 0.208)), accounting for 16.77% of the total effect. The indirect effect on the perceived social support → depression → life satisfaction path was 0.148 (95% CI = (0.088, 0.217)), accounting for 18.38% of the total effect. The indirect effect on the perceived social support → resilience → depression → life satisfaction path was 0.075 (95% CI = (0.041, 0.114)), accounting for 9.32% of the total effect. Additionally, the results showed that the 95% CI corresponding to each path did not contain 0, which indicated that the mediation effect was significant, and the chain mediation was established (Table 4).

## 4. Discussion

This study focused on Chinese medical staff and aimed to explore the mediating roles of resilience and depression on the relationship between perceived social support and life satisfaction. The results indicated that perceived social support was positively correlated with resilience as well as life satisfaction among medical workers in China, but negatively associated with their depressive symptoms. More importantly, the results showed that perceived social support could not only separately impact Chinese medical providers’ life satisfaction through resilience or depression, but could also indirectly affect their life satisfaction via the chain mediation effect of resilience and depression.

### 4.1. The Effect of Perceived Social Support on Medical Staff’s Life Satisfaction

The results of our present study confirmed that perceived social support had a direct and positive impact on life satisfaction among Chinese medical providers. Specifically, medical providers who perceived more sense of care and concern from supportive members tended to be more satisfied with their lives, as compared with those who perceived low to no social support. This finding proved hypothesis 1 and was consistent with previous results with regard to medical workers [9,60] and other general populations [32,34,61]. In addition, the results confirmed that the main effect model of social support theory [62] could be used for explaining the improvement in medical providers’ life satisfaction in the Chinese context. Medical providers who perceived more social support would possess a higher level of understanding and utilize diverse social support resources. In this context, when faced with adverse events, they would be more likely to take a positive action, such as seeking professional help or sharing personal experiences and feelings in social interactions with peers and other supporting members. These positive behaviors help sustain positive emotions, contributing to a sense of achievement and confidence in one’s work [63,64], thus resulting in increasing one’s life satisfaction. In addition, our results also support the bottom-up theory of subjective wellbeing [3]: that high levels of social support could be correlated with domain satisfaction and life satisfaction [65]. Briefly speaking, enhancing perceived social support from family, friends, and significant others is advantageous to the improvement of medical staff’s life satisfaction.

### 4.2. The Mediating Role of Resilience between Perceived Social Support and Life Satisfaction

In line with hypothesis 2, the current study indicated that resilience plays a partial mediating role in the relationship between perceived social support and medical staff’s life satisfaction. That is, perceived social support could not only directly affect life satisfaction, but could also indirectly affect it through psychological resilience, which is consistent with previous studies based on different samples [21,35,46]. Several studies have revealed that perceived social support was positively associated with resilience among medical workers [11,66]. Based on the resilience framework [34], when people face stressors and challenges, external support resources from family, society, and peers can interact with internal psychological resilience factors. For medical professionals, perceived social support can help them cope with psychological tensions, protect them from the adverse effects of various stressful and oppressive events, and improve their social adaptability [67], and thus may make them more resilient in resisting diverse high-level occupational stressors and setbacks. By contrast, when facing stressors and challenges, the lower the external social support resource, the lower the medical staff’s psychological resilience would be, as argued by a more recent study conducted by Xu et al. [66]. Furthermore, the current results also demonstrated that resilience was positively related with medical staff’s life satisfaction. One possible explanation might be that medical workers with high levels of resilience could organize emotional and psychological resources (such as optimism, self-beliefs, and high openness) to cope with adversity or stressful elements successfully in their daily work and lives [68], which in turn leads to high levels of life satisfaction.

### 4.3. The Mediating role of Depression between Perceived Social Support and Life Satisfaction

This study also revealed that depression acts as another mediator between perceived social support and medical professionals’ life satisfaction, which verifies hypothesis 3. For one thing, a substantial body of studies has suggested that perceived social support is negatively correlated with depression [39,40,69]. In particular, with multiple linear regression analysis, a recent study conducted by Fu. et al. [4] found that physicians with a high level of perceived social support were less likely to suffer from depressive symptoms. A similar conclusion could be made in the current study among the Chinese medical staff group. That is, a high level of perceived social support increases medical professionals’ confidence to deal with adversity and stressful events, and thus decreases their risks of being depressed [70]. What is more, these findings are also consistent with the buffering model of social support, which indicates that when confronting high-level life stressors, perceived social support could fully or partially protect individuals from being affected by these stressful events and further enhance their mental health and life satisfaction [62]. Further, the results of this study also proved that depression could significantly and negatively predict life satisfaction in Chinese medical staff, which is in accordance with previous studies among other groups [41,71,72]. Particularly, unlike medical staff in Western nations, Chinese medical work usually involves long-term direct contact with various stressors, such as long working hours, a high intensity of workloads, and doctor–patient conflicts [14,15]. Thus, Chinese staff are more prone to developing negative emotions such as anxiety and depression [11]. If the depression symptoms and other negative emotions are not relieved in time, they may hinder medical staff’s professional performance and influence the healthcare qualities they provide for patients [73], which would in turn pose serious impacts on their satisfaction in life. Above all, medical staff with a higher perception of social support are more capable of decreasing depressive symptoms, which further leads to the improvement in levels of life satisfaction. Hence, interventions on perceived social support may provide practical implications for relieving negative emotions such as depression and anxiety, which indirectly contributes to higher life satisfaction.

### 4.4. The Chain Mediating Effect of Resilience and Depression in Perceived Social Support and Life Satisfaction

In exploring the mechanism between perceived social support and medical staff’s life satisfaction, the current study also shows that the separate mediation effects of resilience and depression on the relationship between perceived social support and life satisfaction are not fully independent. Specifically, the results proved that resilience is not only positively associated with depression; it also could significantly and negatively predict depression. This finding is in accordance with the results of numerous previous studies on diverse populations [40,43,44,45]. In fact, individuals with higher levels of resilience usually have more positive psychological traits (e.g., optimism, serenity, and high openness) and better emotional regulation abilities [30]. Hence, when experiencing multiple stress or adverse circumstances, such as the COVID-19 pandemic, medical staff can actively and quickly adjust their emotional states, so that they can actively and flexibly mobilize internal resources (e.g., resilience) and external resources (e.g., social support) to tackle adversity and stress [74], which in turn leads to reducing depression and other negative emotional experiences [73]. Taken together, medical professionals with high levels of perceived social support tend to have high levels of psychological resilience and possess more positive emotions and adjustment abilities accordingly, which further enable them to more effectively utilize their psychological resources to buffer or reduce the negative emotions induced by adversity, so that they may still experience enough positive emotions and a high sense of life satisfaction, albeit in difficult and stressful situations. Therefore, perceived social support of medical staff may affect their life satisfaction through the chain mediating effect of resilience and depression, which confirms both hypothesis 4 and the compensatory model of resilience [75].

### 4.5. Implications for Clinical Practice

The current study offers both theoretical and practical implications for shedding light on the underlying mechanisms between perceived social support and life satisfaction. Theoretically, this study greatly expands our understanding into the complex interactions among perceived social support, resilience, depression, and life satisfaction for the medical staff group in China. Specifically, though many previous studies have separately identified the crucial roles of perceived social support, resilience, or depression on life satisfaction, few of these have explored their comprehensive effects on individuals’ life satisfaction in the same framework, i.e., simultaneously taking their interactive affects into account. To the best of our knowledge, this study is the first to reveal that perceived social support may indirectly affect individuals’ life satisfaction through the chain mediation effect of resilience and depression.

Practically, the findings in this study have certain practice implications, which may provide valuable guidance on how to implement positive psychological interventions to maintain high levels of life satisfaction for medical staff. According to our findings, it is suggested that hospital administrators should provide diverse resources for social support for their staff. For example, a psychological counseling room may need to be established, which is important for psychologists to build trust relationships with medical staff and provide them with sufficient internal support. Individual case work and group work can also be provided for medical workers to build or reinforce their social networks and acknowledge the roles of different kinds of social support in helping them to alleviate depressive symptoms and enhance their resilience, which could in turn lead to higher levels of life satisfaction. Furthermore, hospital managers and policymakers should develop psychological interventions to promote medical professionals’ resilience at work. For instance, they can arrange regular seminars, conferences, and workshops at the workplace to provide approaches for personal growth for medical staff to enhance their resilience, which can further decrease their depressive symptoms, as proved in this study. Moreover, stress management and structured resilience training programs such as Stress Management and Resiliency Training (SMART) [76] and Mindfulness-Based Stress Reduction (MBSR) [77] can be enacted to prepare medical workers to deal with challenges and difficulties they will encounter. In addition, hospital managers should provide comprehensive counseling and support services for the medical staff as much as possible to help them reduce depressive symptoms. In daily life, more social support should be provided, and depressed staff should be led to perceive more social support to improve their mental state and mood.

### 4.6. Limitations

Our study also has several limitations. First, all data were collected through online self-reported questionnaires. Therefore, it is inevitable that there was a risk of recall bias as well as selection and observation bias, which could further result in inaccurate responses. Second, the research method of a cross-sectional survey utilized in this study cannot confirm the cause–effect relationships between variables. In the future, longitudinal and experimental methods can be used to explore the causality of relationships between perceived social support, resilience, depression, and life satisfaction. Third, we used convenience sampling to collect data from two non-tertiary hospitals in a single city in China, i.e., Shaoguan. However, Shaoguan is a relatively underdeveloped city in both Guangdong Province and in China. In addition, the majority of participants (82.1%) were female, because most of these were recruited from a maternal and child healthcare hospital. Therefore, a normal gender distribution might not be properly represented in our sample. Moreover, since the medical staff participating in this study primarily came from non-tertiary hospitals, their education levels and monthly incomes were generally lower than those working in tertiary hospitals. Therefore, our sample is not fully representative of the entire medical staff group, especially in comparison with those working in more developed and urbanized cities and/or in tertiary hospitals. In summary, we need to be cautious in generalizing our findings because of the biases embedded in the methodological issues mentioned above.

## 5. Conclusions

This research reveals that both perceived social support and resilience have significant predictive effects on medical staff’s life satisfaction, whereas depression might have an adverse effect on their life satisfaction. This study significantly expands our understanding of the underlying mechanisms working in between perceived social support and life satisfaction among China’s medical staff. Moreover, important pathways starting from perceived social support via resilience and depression to life satisfaction further demonstrate the complex interactions among these variables. In addition, the results obtained in this study suggest that enhancing medical staff’s psychological resilience and reducing negative conditions such as depression are promising methods to improve their life satisfaction, which might create a win-win situation for medical staff and their patients.

## Figures and Tables

**Figure 1 ijerph-19-16646-f001:**
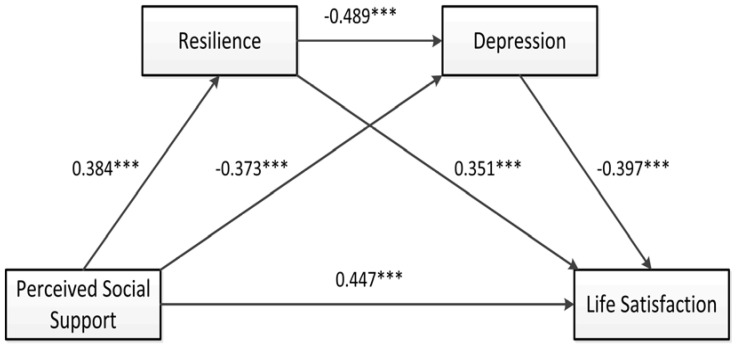
The chain mediation model of perceived social support, resilience, depression, and life satisfaction. *** *p* < 0.001.

**Table 1 ijerph-19-16646-t001:** Socio-demographic characteristics and their differences among primary variables.

Variables	Socio-Demographic Characteristics	*N* (%)	Life Satisfaction	Perceived Social Support	Resilience	Depression
			Mean ± SD	Mean ± SD	Mean ± SD	Mean ± SD
Gender	Male	88 (17.9)	4.41 ± 1.23	5.38 ± 0.97	2.42 ± 0.62	1.17 ± 1.06
	Female	403 (82.1)	4.56 ± 1.26	5.32 ± 1.00	2.32 ± 0.69	1.02 ± 0.92
	*t*		−0.994	0.506	1.194	1.329
	*p* value		0.321	0.613	0.233	0.184
Educational level	College diploma	278 (56.6)	4.64 ± 1.25	5.31 ± 1.03	2.33 ± 0.64	0.98 ± 0.94
	Bachelor’s degree	213 (43.4)	4.39 ± 1.25	5.37 ± 0.94	2.36 ± 0.71	1.15 ± 0.96
	*t*		2.087	−0.674	−0.454	−1.979
	*p* value		0.037	0.500	0.650	0.048
Age	<30	197 (40.1)	4.22 ± 1.29	5.17 ± 1.05	2.27 ± 0.71	1.15 ± 1.05
	31~40	125 (25.5)	4.52 ± 1.13	5.35 ± 0.91	2.26 ± 0.59	1.15 ± 0.87
	41~50	128 (26.1)	4.91 ± 1.21	5.49 ± 1.01	2.50 ± 0.69	0.85 ± 0.84
	51~60	41 (8.3)	4.93 ± 1.20	5.59 ± 0.79	2.41 ± 0.64	0.90 ± 0.89
	*F*		9.893	3.964	3.682	3.305
	*p* value		<0.000	0.009	0.008	0.18
Employment status	Formal employee	301 (61.3)	4.68 ± 1.20	5.43 ± 0.95	2.37 ± 0.64	1.01 ± 0.88
	Contract employee	190 (38.7)	4.31 ± 1.31	5.17 ± 1.04	2.29 ± 0.74	1.12 ± 1.05
	*t*		3.206	2.807	1.211	−1.299
	*p* value		0.001	0.005	0.226	0.195
Marital status	Unmarried	133 (27.1)	4.19 ± 1.27	5.22 ± 1.08	2.25 ± 0.76	1.20 ± 1.10
	Married	347 (70.7)	4.65 ± 1.24	5.37 ± 0.96	2.37 ± 0.65	1.00 ± 0.88
	Divorced or others	11 (2.2)	5.02 ± 1.04	5.49 ± 1.01	2.49 ± 0.69	0.73 ± 0.92
	*F*		7.426	1.244	1.684	2.185
	*p* value		0.001	0.289	0.187	0.132
Position	Doctor	150 (30.6)	4.36 ± 1.19	5.21 ± 0.98	2.25 ± 0.67	1.16 ± 0.89
	Nurse	259 (52.7)	4.60 ± 1.31	5.38 ± 0.99	2.38 ± 0.69	1.03 ± 0.99
	Others	82 (16.7)	4.64 ± 1.17	5.41 ± 1.02	2.37 ± 0.66	0.93 ± 0.89
	*F*		2.121	1.650	1.804	1.713
	*p* value		0.121	0.193	0.166	0.181
Years of working	≤10	238 (48.5)	4.26 ± 1.25	5.18 ± 1.03	2.28 ± 0.70	1.15 ± 1.02
	11~20	98 (20.0)	4.61 ± 1.21	5.46 ± 0.87	2.28 ± 0.61	1.13 ± 0.90
	21~30	117 (23.8)	4.85 ± 1.28	5.48 ± 1.04	2.49 ± 0.67	0.84 ± 0.83
	>30	38 (7.7)	5.06 ± 0.96	5.53 ± 0.72	2.35 ± 0.69	0.88 ± 0.82
	*F*		9.032	4.086	2.79	3.316
	*p* value		<0.001	0.008	0.04	0.02
Professional title	Primary	289 (58.9)	4.39 ± 1.27	5.26 ± 1.02	2.27 ± 0.68	1.14 ± 1.03
	Intermediate	139 (28.3)	4.73 ± 1.15	5.40 ± 0.95	2.39 ± 0.62	0.96 ± 0.82
	Senior	63 (12.8)	4.73 ± 1.35	5.54 ± 0.92	2.54 ± 0.75	0.84 ± 0.79
	*F*		4.349	2.506	4.67	3.514
	*p* value		0.013	0.083	0.01	0.031
Monthly income (RMB)	<3000	44 (9.0)	4.07 ± 1.24	4.91 ± 1.03	2.19 ± 0.75	1.24 ± 1.10
	3000~6000	267 (54.4)	4.41 ± 1.29	5.25 ± 0.99	2.30 ± 0.65	1.12 ± 1.00
	6000~9000	155 (31.5)	4.80 ± 1.16	5.54 ± 0.98	2.44 ± 0.70	0.88 ± 0.80
	>9000	25 (5.1)	4.98 ± 1.15	5.69 ± 0.65	2.49 ± 0.59	0.97 ± 0.91
	*F*		6.435	6.81	2.626	2.813
	*p* value		<0.001	<0.001	0.05	0.039

**Table 2 ijerph-19-16646-t002:** Mean, standard deviation (SD), and correlations for study variables (*N* = 491).

Variables	Mean ± SD	Range	Perceived Social Support	Resilience	Depression	Life Satisfaction
Perceived Social Support	5.33 ± 0.99	1–7	1			
Resilience	2.34 ± 0.68	0–4	0.566 **	1		
Depression	1.05 ± 0.95	0–6	−0.589 **	−0.574 **	1	
Life satisfaction	4.53 ± 1.26	1–7	0.659 **	0.571 **	−0.633 **	1

** *p* < 0.01 (two-tailed test).

**Table 3 ijerph-19-16646-t003:** Regression analysis among variables in the chain intermediary model.

Outcome Variable	Predictor Variable	*R*	*R* ^2^	*F*	*β*	*t*
Resilience	Perceived Social Support	0.578	0.334	34.555	0.384	14.709 ***
Depression	Perceived Social Support	0.663	0.440	47.240	−0.373	−9.261 ***
	Resilience				−0.489	−8.389 ***
Life satisfaction	Perceived Social Support	0.755	0.570	70.754	0.447	8.800 ***
	Resilience				0.351	4.835 ***
	Depression				−0.397	−7.512 ***

*** *p* < 0.001.

**Table 4 ijerph-19-16646-t004:** The chain mediating effect of resilience and depression on the relationship between perceived social support and life satisfaction.

Model Pathways	Effect	Boot SE	95% CI		Relative Mediation Effect %
			Lower	Upper	
Direct effect	0.447	0.051	0.347	0.547	55.53%
perceived social support → resilience → life satisfaction	0.135	0.036	0.070	0.208	16.77%
perceived social support → depression → life satisfaction	0.148	0.033	0.088	0.217	18.38%
perceived social support → resilience → depression → life satisfaction	0.075	0.018	0.041	0.114	9.32%
Total mediation effect	0.358	0.042	0.279	0.442	44.47%

## Data Availability

Due to privacy, the datasets involved in this study are not publicly available, but they are available from the author Nannan Wu on reasonable request.

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
