# Peer review of "The Relationship between Perceived Social Support and Life Satisfaction: The Chain Mediating Effect of Resilience and Depression among Chinese Medical Staff"

_ijerph, 2022, doi:10.3390/ijerph192416646_

Round 1
Reviewer 1 Report
Dear Authors,
The proposed research work is exploratory in nature. The limitations specify those methodological issues that prevent the generalisation of the results. This is the main handicap of the contribution.
As strengths, it is worth highlighting the important evidence that supports both the studies and that accompanies the discussion of the results. It should also be noted that the methodology is adequate and the results are clearly presented.
In line 86 it would be advisable to reword "Therefore, as in [20,22]" for better understanding.
Reviewer 2 Report
Thank you for inviting me to review the interesting paper "The Relationship Between Perceived Social Support and Life Satisfaction: The Chain Mediating Effect of Resilience and Depression among Chinese Medical Staff " submitted to the editorial board of the journal International Journal of Environmental Research and Public Health. Overall, the paper submitted for review fits the scope of the journal and may be of interest to readers, given the journal's profile.
The presented cross-sectional study is properly designed and justified and was designed to explore the mechanism underlying the influence of perceived social support on medical staff's life satisfaction. A quite large sample consisting of 533 medical staff (n=150 doctors, n=259 nurses, and 82 others) completed the Multidimensional Scale of Perceived Social Support, the Satisfaction with Life Scale, the Connor and Davidson Resilience Scale 19 and the Depression subscale of Depression, Anxiety and Stress Scales (DASS-21).
The introduction is a very organized and comprehensive source of knowledge and introduces the reader to the issue addressed in the course of the authors' own research. In my opinion, the content itself could have been shorter due to the research rather than the review nature of the paper, not me; however, I leave this to the Editor's discretion. Subsections 1.3. and 1.4. address similar content and could be combined and shortened.
The research questions are original and well-defined and could translate into advances in existing knowledge on topics related to perceived social support on medical staff's life satisfaction.
Material and method show the major subsections typical of research papers, making it easier to navigate throughout the text. Consideration could be given to introducing inclusion and exclusion criteria, as well as presenting a STROBE flow chart at each stage of qualifying study participants.
The results are appropriately interpreted and provide the significant scientific potential for researchers, as well as health care professionals and policymakers.
All the conclusions in the paper are justified and supported by the results of the authors' research based on well-constructed research hypotheses.
The data presented are clear and valid and analyzed based on appropriate statistical tests, which correspond to the standards of analysis and presentation of results.
The conclusions are interesting and directly respond to the research hypotheses, corresponding to the work assumptions in an unobjectionable manner.
The manuscript does not contain an excessive number of self-citations. The cited references are mostly recent publications. Nevertheless, quite a few of them are older than 5-10 years and should be refreshed. Some are even from the 1980s and 1990s.
Figures 1 and Tables 1-3 are adequate and properly present the data, making them friendly for interpretation and understanding. Figure 1 presents the chain-mediation model that deserves special recognition.
The ethical and data availability statements are not questionable and, in my opinion, are properly prepared. There are clear statements that the study was conducted in accordance with the Declaration of Helsinki, and approved by the Ethics Committee of Medical School of Shaoguan University (IRB number: yxyllscb202202) as well as informed Consent Statement: Informed consent was obtained from all subjects involved in the study. In "Funding: Please add: This research..." please remove "Please add".
For me, the English language is adequate and understandable, while I am not a native speaker so I ask the language editor of the editorial board to proofread the entire text before accepting it for publication.
In summary, the entire article is written appropriately, and well-organized, its structure is not objectionable. The manuscript is clear, and relevant to the field of public health. Is the manuscript scientifically sound and coherent, and the study design is adequate to verify the hypotheses presented. The entire submission was prepared with due diligence and in accordance with the journal's formal guidelines for authors.
My recommendation is: Accept after minor revisions.
